Morphology and vocalization comparison of the Houston Toad and the Dwarf American Toad: implications for their historic range

MacLaren Andrew R. amaclaren89@gmail.com 1
Hibbitts Toby J. 2 3
Forstner Michael R.J. 4
McCracken Shawn F. 5
1 Cambrian Environmental , Austin , TX , United States of America
2 Biological Research and Teaching Collection, Department of Ecology and Conservation Biology, Texas A&M University , College Station , TX , United States of America
3 Texas A&M Natural Resources Institute , College Station , TX , United States of America
4 Department of Biology, Texas State University , San Marcos , TX , United States of America
5 Department of Life Sciences, Texas A&M University - Corpus Christi , Corpus Christ , TX , United States of America
De Baets Kenneth
Electronic publication date: 2024 Jul 8
Publication date: 2024
Volume: 12
Electronic Location ID: e17635
Received 2023 Oct 11; Accepted 2024 Jun 4
Copyright: ©2024 MacLaren et al.
Copyright year: 2024
Copyright holder: MacLaren et al.
License: This is an open access article distributed under the terms of the Creative Commons Attribution License, which permits unrestricted use, distribution, reproduction and adaptation in any medium and for any purpose provided that it is properly attributed. For attribution, the original author(s), title, publication source (PeerJ) and either DOI or URL of the article must be cited.
License URL: https://creativecommons.org/licenses/by/4.0/

Keywords: Endangered species, Bufonid, Vocalization, Comparative Morphology, Houston Toad, Dwarf American Toad, Museum collections, Call description, Acoustics, Dichotomous keys

Funding: The authors received no funding for this work.

==============================
Documenting changes in the distribution and abundance of a given taxon requires historical data. In the absence of long-term monitoring data collected throughout the range of a taxon, conservation biologists often rely on preserved museum specimens to determine the past or present, putative geographic distribution. Distributional data for the Houston Toad (Anaxyrus houstonensis) has consistently been confounded by similarities with a sympatric congener, the Dwarf American Toad (A. americanus charlesmithi), both in monitoring data derived from chorusing surveys, and in historical data via museum specimens. In this case, misidentification can have unintended impacts on conservation efforts, where the Houston Toad is federally endangered, and the Dwarf American Toad is of least concern. Previously published reports have compared these two taxon on the basis of their male advertisement call and morphological appearance, often with the goal of using these characters to substantiate their taxonomic status prior to the advent of DNA sequencing technology. However, numerous studies report findings that contradict one another, and no consensus on the true differences or similarities can be drawn. Here, we use contemporary recordings of wild populations of each taxon to test for quantifiable differences in male advertisement call. Additionally, we quantitatively examine a subset of vouchered museum specimens representing each taxon to test previously reported differentiating morphometric characters used to distinguish among other Bufonids of East-Central Texas, USA. Finally, we assemble and qualitatively evaluate a database of photographs representing catalogued museum vouchers for each taxon to determine if their previously documented historic ranges may be larger than are currently accepted. Our findings reveal quantifiable differences between two allopatric congeners with respect to their male advertisement call, whereas we found similarities among their detailed morphology. Additionally, we report on the existence of additional, historically overlooked, museum records for the Houston Toad in the context of its putative historic range, and discuss errors associated with the curation of these specimens whose identity and nomenclature have not been consistent through time. These results bookend decades of disagreement regarding the morphology, voice, and historic distribution of these taxa, and alert practitioners of conservation efforts for the Houston Toad to previously unreported locations of occurrence.

Introduction

Vertebrates are currently experiencing range contractions and population declines globally, contributing to what has been designated as our sixth mass extinction event (Ceballos, Ehrlich & Dirzo, 2017). Among declining vertebrates, amphibians are one of the groups reported to be most at risk (Wake & Vredenburg, 2008; Bishop et al., 2012). In response to these steep declines, nearly 300 species have been proposed as candidates for protection under the U. S. Endangered Species Act (U. S. Fish and Wildlife Service, 1973, as amended). Despite an effort by researchers to place species on the endangered list, a recent study found that approximately 13% of candidate species possessed any kind of pre-listing conservation plan (BenDor & Riggsbee, 2017). The basis of listing decisions for candidate species is under a directive to follow the best available science (U. S. Fish and Wildlife Service, 1973, as amended), but for many species these data are not available; agencies often demonstrate a preference for indeterminate findings regarding candidate species (Murphy & Weiland, 2016).

Conservation biologists rely on historical data to document changes in the distribution and abundance of species (Skelly et al., 2003). Long-term monitoring is the most ideal way to document these changes, but these data do not exist for most species of concern (e.g., candidate species for protection under the U.S. Endangered Species Act). In the absence of these data, existing information on the historical distribution of a species may be gleaned from natural history collections, which are the only verifiable source of species presence at a given time and place. For species of conservation concern, these data are routinely collated and compared to contemporary observations, describing expansion or contraction in geographic distribution (Shaffer, Fisher & Davidson, 1998). Surprisingly, for the United States’ first federally protected amphibian, the Houston Toad (Anaxyrus houstonensis; (Gottschalk, 1970; Sanders, 1953), these data have not been examined.

The Houston Toad (hereafter HT) is a diminutive member of the A. americanus complex, occurring in East-Central Texas, USA (Sanders, 1953; Masta et al., 2002; Sirsi, Rodriguez & Forstner, 2024). The adults show a preference for deep sandy soils where forest or woodland cover occurs proximal to breeding ponds (Potter et al., 1984). In its description, the species’ range is hypothesized to extend north and east to Arkansas and Oklahoma, following the occurrence of these features (Sanders, 1953). However, the eastern limit for this species’ distribution has been speculated to occur at the Trinity River in Texas, citing the lack of detections beyond this perceived barrier (Seal, 1994). The known range of HT includes counties where toads are detected annually through auditory surveys and incidental encounters (e.g., Bastrop & Robertson counties), counties where only a single or few detections have been documented (e.g., Freestone & Brazos counties), and localities where extirpation of the species has been reported (e.g., Liberty, Harris, and Fort Bend counties; Brown, 1975; Sanders, 1953; Yantiss & Price, 1993). Long-term monitoring for the species has taken place within occupied counties (i.e., harboring well known populations) only, but monitoring occurs opportunistically within adjacent counties where detections have historically been rare.

In North America, Toads (family Bufonidae) are generally distinguished from one another based on morphological differences in the arrangement and shape of their parotoid glands and cranial crests (flesh-covered bony protuberances adorning the skull), coloration or skin pattern (when alive, as opposed to in preserved specimens; Tipton et al., 2012), as well as behavior (especially male advertisement call; Masta et al., 2002; Fontenot, Makowsky & Chippindale, 2011). HT was described prior to the establishment of modern conventions for the description of a new species (Sanders, 1953; Winston, 1999), and the primary evidence presented by Sanders (1953) are qualitative skeletal features. While the work of Sanders (1953) provides much of the first general information about the species, it does not provide us with an assessment of HT morphology, natural history, or male advertisement call relative to its nearest relatives, as we would expect in any modern anuran description. The nearest relative to HT is the Dwarf American Toad (A. americanus charlesmithi; DAT hereafter). This species was described just one year after HT (Bragg, 1954), and similarly, its description was without a proper treatment of quantified differentiation relative to nearby congeners, even failing to acknowledge the preceding description of HT entirely. Historically, DAT was thought to be excluded from Texas via the Red River, occurring only in Oklahoma (Sanders, 1953), but this has since been disproven (Dixon, 2013). Putatively, these species are allopatric within Texas. Morphologically, HT has traditionally been distinguished from DAT by its enlarged cranial features and warts on the hind limbs (Sanders, 1953). However, these features are highly plastic in natural populations, and it has been suggested that this differentiation between species can potentially be contributed to the holotype of HT being unusually large (Seal, 1994). It has also been reported that male advertisement call of HT and DAT are indistinguishable (R. A. Thomas, H. C. Dessauer, 1982, unpublished data; Preliminary report: Taxonomy of the Houston Toad, Bufo houstonensis Sanders). Although the foundational literature contains disagreement on this point (Brown, 1973).

Research utilizing molecular methods (i.e., DNA sequencing) to investigate the phylogenetic relationships within the A. americanus complex has not arrived at a singular topology that describes the species thought to comprise this group (Pauly, Hillis & Cannatella, 2004; Goebel, Ranker & Olmstead, 2009). Toads within this group lack reproductive barriers to hybridization, and thus similarities in phenotype, as well as male advertisement call, have been shown to not correlate with evolutionary relationships (Fontenot, Makowsky & Chippindale, 2011), suggesting that similarities in advertisement call evolved independently (Masta et al., 2002). Recent molecular research substantiates that HT and DAT are distinct, while acknowledging secondary contact (i.e., gene flow) between geographically proximal populations of HT and DAT (Sirsi, Rodriguez & Forstner, 2024). These findings highlight that there are areas where these toads have previously co-occurred within Texas.

Many studies have shown that hybridization among sympatric Bufonids is common (Strecker, 1915; Blair, 1963; Ballinger, 1966; Brown, 1971; Hillis, Hillis & Martin, 1984; Masta et al., 2002; Chivers, 2016; Sirsi, Rodriguez & Forstner, 2024). This can result in shared morphological characters that confuse or undermine identification at the species level. Many contradictory opinions exist within the primary literature with respect to the morphology of these animals. For example, in its initial description, Sanders (1953) describes the parotoid glands of HT as “diverging posteriorly”, and Bragg (1954) describes those of DAT as “nearest together in the middle”. Nevertheless, contemporary dichotomous keys (Tipton et al., 2012; Dixon, 2013) report that both possess parotoid glands that are nearest together anteriorly and that ultimately HT are differentiable from DAT by the presence of enlarged warts (which occur on the tibia of the latter and enlarged post-orbital crests among the former). Hillis, Hillis & Martin (1984) provided a methodology using quantitative morphometrics to diagnose species assignment as well as hybrid status for HT and two sympatric species (previously considered congeners) Woodhouse’s Toad (A. woodhousii) and the Coastal Plains Toad (Incilius nebulifer), but these methods have not been utilized more broadly within the A. americanus complex. Inconsistencies in reported phenotype (in morphology or vocalization) can lead to misidentification in the field (or in collections), which may have produced unintended negative impacts on conservation efforts, where HT is federally endangered, and DAT is of least concern.

In response to decades of disagreement within foundational literature (i.e., species descriptions, dichotomous keys, state and federal agency reports) with respect to the similarities in morphology, male advertisement call, and historic range of these taxa, we sought to summarize and examine disparities within previous research by determining whether HT and DAT (1) are morphologically distinct, (2) differ on the basis of male advertisement vocalization, and (3) historically occurred in a larger geographic area than previously reported. We use three unique datasets to examine these questions, which include quantitative morphological measurements (i.e., morphometrics) taken from formalin fixed specimens, audio recordings of males chorusing, and photos representing museum vouchers of each putative species reported to have been collected outside their known historic range. To the best of our knowledge, no study has examined or compared the morphology of sympatric toads within Texas, with the stated goal of identifying reliable differentiating features for use in a field setting. Thus, the features used in discriminating among species (e.g., within dichotomous keys) are at best qualitative, and perhaps erroneous. Portions of this text were previously published as part of a dissertation (https://digital.library.txst.edu/items/b664caaa-9f83-47ca-923b-b9aeefb48455/full).

Materials & Methods

Morphometric comparison

We sought to replicate and expand the methodology of Hillis, Hillis & Martin (1984) to identify quantitative differences between HT and DAT, as well as to test the effectiveness of this methodology when applied to preserved specimens. We accessed vouchers housed at the Biological Research and Teaching Collection at Texas A&M University, College Station, Texas, as well as the herpetological teaching collection at Texas State University, San Marcos, Texas. Unique catalogue numbers for each specimen measured, if available, are provided within our supplementary dataset. We measured vouchers of HT (n = 51), DAT (n = 50), and the Coastal Plains Toad (n = 39) (referred to as the Gulf Coast Toad, Bufo valliceps, by Hillis, Hillis & Martin, 1984; see Frost, Mendelson III & Pramuk, 2009) as a control since they are morphologically distinct. All specimens measured for this dataset are housed at the Biological Research and Teaching Collection at Texas A&M University, College Station, TX, USA. Following Hillis, Hillis & Martin (1984), we measured the following eight features (Fig. 1): head width (HW; at widest part of head), distance between interocular crests (DBIC; at widest point), mean parotoid length (MPL; horizontal length of each parotoid gland), mean parotoid width (MPW; maximum width of parotoid glands), mean length of tibiofibula (MTFL; distance between knee and top of foot), snout-urostyle length (SUL; tip of snout to posterior end of urostyle), mean distance between parotoid and postorbital ridge (MPPG; the gap between anterior of parotoid and the postorbital cranial feature, at its narrowest point), and mean thickness of postorbital ridge (MPT; measured at its widest point anteriorly to posteriorly). Measurements of characteristics present on both the right and left side were averaged to account for lateral asymmetry (Fig. 1). All morphological characters were measured by A. R. MacLaren using digital calipers (Mitutoyo Corporation, Kawasaki, Kanagawa, Japan) to a precision of 0.01 mm. To control for the effect of body size (i.e., snout-urostyle length) for each specimen, we performed residuals analyses using ordinary least-squares regression. We conducted linear discriminant function analyses (LDA; package “MASS”; Ripley et al., 2013; R Core Team, 2024) to visualize size-adjusted morphological characters that may discriminate between species. We treated species as a priori groups to test for quantifiable differences between HT and DAT after preservation. To test for collinearity, we calculated pair-wise Pearson’s correlation coefficient for all morphological characters.

Figure 1 Simplified dorsal view of a Houston Toad (Anaxyrus houstonensis) demonstrating morphometric measurements taken.

Metrics numbered 1–8 correspond to: (1) head width, (2) distance between interocular crests, (3) parotoid length, (4) parotoid width, (5) length of tibiofibula, (6) snout to urostyle length, (7) distance between parotoid and postorbital ridge, (8) thickness of postorbital ridge. Simplified illustration traced by A.R. MacLaren from photograph by S.F. McCracken.

Advertisement call comparison

We also examined advertisement call structure of HT and DAT. HT vocalizations were collected from two sites, each within a different county (Bastrop and Robertson Counties, Texas, USA). DAT vocalizations were collected in Cleveland County, Oklahoma, USA. Access to field sites in Robertson County, Texas, were granted by Laredo Petroleum to Michael R.J. Forstner. Access to field sites (the Griffith League Ranch) in Bastrop County Texas was granted by the Boy Scouts of America Capitol Area Council to Michael R.J. Forstner. Audio Recordings made by C. McAllister in Oklahoma occurred on public property. No protected areas were accessed during this study; however, we omit exact locations for the purpose of protecting sensitive populations.

In Texas, we recorded vocalizations using a SongMeter SM3 (Wildlife Acoustics, Maynard MA, USA). Deployment of these devices follows the methods outlined in MacLaren, McCracken & Forstner (2018). In Oklahoma, we recorded calls using a hand-held digital recording device (Tascam DR05; Montabella CA, USA), which was oriented in the direction of the chorusing individuals. As is the case with other remote acoustic monitoring efforts, it is difficult or even impossible to determine how many individuals can be heard in any given recording. Therefore, we were unable to control for variation at the individual level (Moriarty & Cannatella, 2004). Likewise, in Oklahoma, although the recording device is handheld, not all chorusing animals were visually observed, or able to be counted in a systematic manner. Passive recordings were collected with permission from landowners, but did not require any formal permitting from state, federal, or local authorities, and we did not capture any incidental recordings of non-participating human subjects that may have been unaware of the presence of our recording devices. Ultimately, we collated and described 50 HT and 49 DAT vocalizations from an unknown number of individuals at each location. We extracted characteristics from each vocalization in Raven Pro (V.1.5, Cornell University; Bioacoustics Research Program, 2014). These characteristics include call dominant frequency (DF; frequency in Hz at peak call amplitude), call length (CL; duration in seconds from first call pulse to last call pulse), pulse number (PN; the number of pulses in each call divided by the call length), and frequency range (FR; the number of Hz between lowest and highest frequency). We then used these characters to estimate whether each call is narrow or broad in its frequency breadth. We explored patterns in advertisement call data by creating univariate box and whisker plots of raw data as well as performing t-tests for each variable to test for differences in call characteristics between the two species (α = 0.05, package “stats”; R Core Team, 2024).

Figure 2 Map demonstrating the range of the Houston Toad and Dwarf American Toad in East Central Texas, USA, according to varying information sources.

Counties with bold orange outline appear among primary literature as those historically inhabited by the Houston Toad (A houstonensis). Counties reported to be the site of collection for Houston Toads within natural history collections are highlighted in green. Counties reported to be the site of collection for Dwarf American Toads (A. a. charlesmithi) within natural history collections are highlighted in light orange. Black stars or filled circles reflect reports of Houston Toad and Dwarf American Toad vouchers within (Dixon, 2013), respectively. Map created using ArcGIS.

Unresolved museum records

We located museum vouchers for HT and DAT through VertNet (vertnet.org), a collaborative and open-access biodiversity data portal that includes specimen collections records. We queried VertNet for three epithets: “houstonensis”, “americanus”, and “terrestris”. We included “terrestris” in an effort to ensure all previously proposed nomenclature for the DAT were represented (Bragg, 1954). We requested photos of the dorsal, ventral, and cranial features for each vouchered specimen reported by VertNet to have been collected outside of the known range for these species (n = 30, Fig. 2). We qualitatively evaluated these photographs to determine whether specimens had been misidentified at the time of collection or accession into museum collections based on the presence or absence of diagnostic morphological characters, using the dichotomous keys available from Tipton et al. (2012), an adaptation of Dixon (2000), updated as Dixon (2013) and Powell, Collins & Hooper (2012). We qualified photos based on whether specimens possessed parotoid glands closest at midpoint (A. woodhousii), versus closest anteriorly (HT and DAT); whether warts of the tibia were larger than those of the thigh (DAT), or uniform in size on both tibia and thigh (HT); whether post-orbital cranial crests were enlarged relative to the inter-orbitals (HT), or equal in size to the inter-orbitals (DAT); whether the chest is spotted to entirely black (A. americanus), or pale (A. woodhousii complex); the number of dorsal warts encircled in dark pigment (1–3 warts = A. americanus; up to 6 warts = A. woodhousii complex). According to Powell, Collins & Hooper (2012), HT can be differentiated from all other toads in the genus Anaxyrus by examining their post-orbital cranial crests, which are thicker than their inter-orbital cranial crests. It is important to note that many of these dichotomous couplets are examined by Hillis, Hillis & Martin (1984), which we replicate in the above sections. In this section, we focused on the use of dichotomous characters in the absence of reliable information relating to specimen size (required for the previous section), which was not available to us within our photographic data. Specimens that have been formalin fixed and stored in ethanol often show signs of shrinking, decalcification of bone, and darkening or fading of pigmentations (Simmons, 1995). Thus, many of the characters used to differentiate North American Bufonids may be absent or indistinguishable, obscuring our ability to reliably identify toads from photographs. For most evaluated specimens, sex was not provided by the collectors nor able to be determined in the photograph. Thus, we did not include sex within our qualitative analyses.

Results

Our attempt to replicate and expand the morphometric comparison performed by Hillis, Hillis & Martin (1984) revealed remarkable similarity between HT and DAT (Fig. 3). Morphometric differences between HT and the Coastal Plains Toad agree with Hillis, Hillis & Martin (1984), indicating that we successfully followed their methods, and further supporting the conclusion that HT and DAT are similar morphometrically. The previous authors did not account for body size, except by removing it from the analysis entirely through step wise procedures. In our study, after conducting regression analyses, the remaining seven morphometric characters were all highly correlated with one another (Table 1).

Figure 3 Plot of first two axes of a linear discriminant function analysis using measurements of eight morphological characters among Coastal Plains Toads (I. nebulifer; n = 39), Dwarf American Toads (A. a. charlesmithi; n = 50), and Houston Toads (A. houstonensis; n = 51).

Table 1 Matrix of pairwise Pearson’s correlation coefficient for size-adjusted morphological characters among A. houstonensis and A. a. charlesmithi.

HW, head width; DBIC, Distance between interocular crests; MPL/W, Mean parotoid length/width; MTFL, mean tibiofibular length; MPPG, mean parotoid to postorbital gap; MPT, mean postorbital thickness.

	HW	DBIC	MPL	MPW	MTFL	MPPG	MPT	
HW	1	0.142	0.628	0.343	0.476	0.476	0.476	
DBIC	–	1	0.860	0.979	0.938	0.938	0.938	
MPL	–	–	1	0.946	0.983	0.983	0.983	
MPW	–	–	–	1	0.989	0.989	0.989	
MTFL	–	–	–	–	1	1	1	
MPPG	–	–	–	–	–	1	1	
MPT	–	–	–	–	–	–	1	

Male advertisement call data indicate that HT vocalizations differ from DAT with respect to dominant frequency and frequency range, but do not differ in call length or pulse number. We found the mean dominant frequency for HT to be 2,034 ± 63 Hz, and 1,892 ± 154 Hz for DAT (p < 0.001). Similarly, mean frequency range was 565.0 Hz and 708.8 Hz (p < 0.001) for HT and DAT, respectively (Table 2).

Table 2 Summary statistics of call characteristics for the calls of Houston Toad (A. houstonensis) and the Dwarf American Toad (A. a. charlesmithi).

CL, Call length; FF, Frequency Range; PN, Pulses per second; DF, Dominant Frequency.

Dwarf American Toad	
	Min	Mean	Median	Max	
CL (s)	2.66	12.65	12.97	26.325	
FR (Hz)	363.8	708.85	691.2	1,247.6	
PN	21.5	26.64	26.57	34	
DF (Hz)	1,050	1,891.7	1,894.9	2,239.5	
Houston Toad	
	Min	Mean	Median	Max	
CL (s)	2.61	11.14	10.782	18.86	
FR (Hz)	269.8	564.97	582.8	1,088.4	
PN	22	27.07	27.25	31.5	
DF (Hz)	1,875	2,033.95	2,046.85	2,125	

For both species, unresolved museum records were discovered within a portion of the range reported by Dixon (2013), as well as beyond that range (Fig. 2). Qualitative assessment of visible morphological features was not easily achieved, and as the results of our previous morphometric analysis revealed, many of these traits fail to differentiate between HT and DAT. We believe that most museum vouchers putatively identified as DAT actually represent Woodhouse’s Toad (A. woodhousii), a more common and widely dispersed toad in Texas, based on the location of their parotoid glands (closest to each other at their midpoint versus anteriorly for HT and DAT) and their warts per dorsal spot not exceeding three (Table 3; Powell, Collins & Hooper, 2012; Dixon, 2013).

Table 3 Qualitative assessment of the morphological features present on unresolved museum vouchers.

A priori indicates the epithet under which each specimen is catalogued, parotoid indicates the point at which the two swollen glands of the toad are nearest to one another, crest size indicates whether the supraorbital and postorbital are uniform in thickness or the postorbital are enlarged, chest pattern is the degree to which the venter of each toad is colored, warts per spot indicates the number of large dorsal warts encircled by black pigment. Using these characteristics, we provide a qualitative assessment of the likely identity of these specimens, as well as the county, year, museum, and specimen number taken from each catalogued record. N/A indicates a lack of photographs demonstrating this feature.

A priori	Parotoid	Wart size	Crest size	Chest pattern	Warts per spot	Qualitatively	County	Year	Museum	Specimen no.	
americanus	middle	n/a	uniform	black throat	n/a	woodhousii	Chambers	1955	BYU	38773	
americanus	–	–	–	–	–	juvenile	Dallas	1928	USNM	75355-56	
americanus	–	–	–	–	–	juvenile	Grimes#	–	OMNH	21728	
americanus	middle	enlarged	enlarged	spotted	1	houstonensis	Harris	1952	SDNHM	42045	
americanus	middle	uniform	uniform	pale	2	woodhousii	Harris	1966	UAMZ	A1535	
americanus	middle	enlarged	enlarged	pale	7	woodhousii	Hunt	1957	UTEP	14438	
americanus	anteriorly	enlarged	uniform	flecks	6	woodhousii	Nacogdoches	1968	TNHC	79899	
americanus	–	–	–	–	–	cleared and stained	Orange	1968	YPM	6842	
americanus	middle	uniform	uniform	spotted	2	houstonensis/ americanus	Sabine	1934	USNM	99771	
americanus	anteriorly	uniform	uniform	spotted	1	houstonensis/ americanus	Panola	1961	KU	70013	
americanus	anteriorly	uniform	uniform	black throat	4	woodhousii	Smith	2000	KU	289496	
americanus	anteriorly	enlarged	uniform	black throat	0	woodhousii	Smith	2000	KU	289469	
americanus	anteriorly	enlarged	uniform	spotted	1	americanus	Smith	2001	KU	289499	
houstonensis	middle	uniform	uniform	spotted	1	houstonensis	Brazos	1958	MSUM	n/a	
houstonensis	middle	uniform	uniform	black throat	3	woodhousii	Erath	1967	ASNHC	14667	
houstonensis	middle	enlarged	enlarged	black throat	0	woodhousii	Houston	1967-68	ASNHC	14657	
houstonensis	middle	n/a	uniform	black throat	1	woodhousii	Houston	1967-68	ASNHC	14668	
houstonensis	middle	enlarged	uniform	black throat	3	woodhousii	Houston	1967-68	ASNHC	14671	
houstonensis	middle	enlarged	uniform	pale	2	woodhousii	Houston	1967-68	ASNHC	14672	
houstonensis	middle	uniform	enlarged	spotted	1	houstonensis	Houston*	1959	LSUMZ	9309	
houstonensis	middle	uniform	uniform	spotted	2	houstonensis	La Calcasieuˆ	1969	LSUMZ	47849	
houstonensis	middle	uniform	uniform	flecks	5	woodhousii	Travis	–	UTA	42188	
houstonensis	middle	uniform	uniform	flecks	5	woodhousii	Travis	–	UTA	41629	
houstonensis	middle	n/a	uniform	n/a	3	woodhousii	Trinity	–	UTA	42438	
houstonensis	middle	uniform	uniform	n/a	2	woodhousii	Walker	–	UTA	40636	
houstonensis	middle	n/a	uniform	n/a	0	woodhousii	Walker	–	UTA	41633	
houstonensis	middle	n/a	enlarged	n/a	1	woodhousii	Walker	–	UTA	41634	
houstonensis	middle	uniform	enlarged	n/a	n/a	woodhousii	Walker	–	UTA	40638	
houstonensis	middle	n/a	uniform	n/a	6	woodhousii	Walker	–	UTA	40637	
houstonensis	–	–	–	–	–	juvenile	Harrison	1972	OSUM	n/a	
terrestris	middle	uniform	uniform	black throat	1	houstonensis	Leon	1945	FMNH	46795	
Notes.

# Specimen verified as A. houstonensis by A. Bragg.

* Specimen tag reads “Houston, Texas” indicating the locality of Houston County is in error.

ˆ Specimen tag reads “Texas, Sam Houston State Park” creating confusion over the exact location of collection.

Specimens recovered from Sabine, Panola, and Smith counties have seemingly been correctly identified as DAT (Table 3). Collected in the 1930s, 1960s and 2000s, respectively, these records indicate that the range of DAT might presently, or previously extend further south into Texas following the Sabine River (Fig. 2). Similarly, museum vouchers for HT were mostly misidentified by collectors in the field or at the time of accession into museum collections. Thirteen of the 18 specimen photographs that we reviewed possessed qualitative traits of Woodhouse’s Toad, generally lacking enlarged post-orbitals and possessing parotoid glands nearest at their midpoint. Specimens recovered for counties already known to currently, or historically, contain populations of HT were correctly identified as such (Table 3, Fig. 2). We discovered a single meaningful voucher able to be discerned as a preserved HT originating from outside its previously known range, which expands its historic distribution to include Brazos County, Texas (MacLaren & Forstner, 2017).

Multiple specimens we sought to verify returned as photographs of juvenile animals, lacking many of the morphological features used to differentiate among these species (Table 3). A toad collected from Orange County, Texas, in 1968 was reportedly a voucher of DAT, but has since been cleared and stained, a preparation that also prevented our ability to qualify the species identity due to lack of visible morphology (e.g., wart pigmentation).

Discussion

Our attempt to expand the research of Hillis, Hillis & Martin (1984) to include DAT failed to highlight any meaningful quantifiable morphometric difference between HT and DAT. This may indicate several things. First, HT and DAT may truly be morphologically indistinguishable. Second, the measured morphometric characters may simply not capture the signal of morphological differentiation present among the specimens considered here. Third, the use of a univariate measure of body size may be inappropriate and perhaps future research should be expanded to include a multivariate consideration (McCoy et al., 2006). These findings imply that the morphological features traditionally used to differentiate among Bufonids of the americanus complex may not be reliable, and this symptom may be exacerbated among preserved vouchered specimen. As we have shown here, the physical frame of a generalized Bufonid can easily be quantified using digital calipers and quickly compared among its allies. In many cases, these differences in pattern and coloration are used as differentiating features (Dixon, 2013), but these features are seldom retained once preserved. This problem is also compounded as specimens age, leaving our oldest historical vouchers with the fewest distinguishing characters. It may be the case that the morphological proportions of HT and DAT differ, in life or preserved, and that the seven features retained in our discriminant analysis were unable to observe this signal. Future research utilizing advanced methods of morphometric analysis, such as expanding to a greater number of landmarks, or machine learning techniques (MacLeod, 2017), to investigate other aspects of these animals’ morphology could better determine if true differences exist, and if so, how these differences can be reliably detected in a field setting.

Measuring anuran advertisement calls to infer phylogenetic relationships, or hybrid status, had not become commonplace until after HT and DAT were described (Zweifel, 1968; Cocroft & Ryan, 1995). Brown (1973) reports that differences in advertisement call are dramatic, primarily pulse number (32.2 for HT and 48.3 for American Toad), inferred from eleven HT calls compared to the extrapolated results of Zweifel (1968) for American Toad. Ultimately, Brown (1973) concludes that for this reason HT cannot be a subspecies as had been suggested by Blair (1957). Likely due to attention following the federal listing of HT as endangered, Thomas & Dessauer re-examined this question (R. A. Thomas, H. C. Dessauer, 1982, unpublished data; Preliminary report: Taxonomy of the Houston Toad, Bufo houstonensis Sanders), stating that the findings of Brown (1973) were intentionally biased, citing that he compared HT calls with those of American Toads (A. americanus) from New Jersey, rather than more nearby populations of DAT (a subspecies of American Toad). When Thomas & Dessauer compared the calls of HT (n = 2) to DAT from Oklahoma (n = 2), they reported no differences between the advertisement calls of these two congeners (R. A. Thomas, H. C. Dessauer, 1982, unpublished data). However, our findings reveal some differences, mainly that DAT calls have a lower dominant frequency compared to HT. While we detected significant differences at a robust sample size, if we consider our findings in the broader context of advertisement calls within the family Bufonidae, they are likely to be very similar (Fontenot, Makowsky & Chippindale, 2011; Table 2, Fig. 4).

Figure 4 Boxplot of raw male advertisement call character data for the Houston Toad and Dwarf American Toad.

(DF, Dominant Frequency (Hz); CL, Call length (seconds); FR, Frequency Range (Hz); and PN, Pulses per second). Values for DF and FR differ significantly (p < 0.001).

Other mechanisms could explain variation in call characters. Pulse number and dominant frequency can decrease with lower body temperatures (Zweifel, 1968). We were unable to measure body temperature in our study, although average air temperature varied by less than 3 °C during recording times. Additionally, we have routinely observed HT choruses in which increased variation in dominant frequency occurs due to male-male competition (A.R. MacLaren pers. obs., Fig. 5); variation at such limited scale is unlikely to carry any phylogenetic signal, as has been suggested in the past (Zweifel, 1968; Cocroft & Ryan, 1995).

Figure 5 Spectrogram (i.e., frequency kHz over time in seconds) of Houston Toad (A. houstonensis) vocalizations collected within Bastrop County, Texas, USA.

When multiple individuals are present advertisement calls vary in frequency (kHz; Y-axis) demonstrating behavioral modulation of this characteristic. Spectrogram generated in Kaleidoscope 4.3.1 (Wildlife Acoustics). Audio visualized in this figure was not used within our analysis of advertisement call comparison.

Through qualitative review of specimen photographs, we found that morphological structures proposed as unique to HT (i.e., enlarged postorbital crests) also occur in toads outside their putative range; conversely, we found toads within this range that possess features putatively unique to DAT (i.e., enlarged tibial warts). Nearly all toads vouchered as either HT or DAT possessed parotoid glands nearest together at their midpoint in contradiction with contemporary dichotomous keys (Tipton et al., 2012; Powell, Collins & Hooper, 2012; Dixon, 2013). In life, HT have an overall darker appearance than their sympatric congeners; however, due to variability in collection and procedures or reagents used to preserve the specimens, the overall pattern or color was uninformative to this study.

We discovered museum records catalogued as HT collected outside the putative range described for this species (Fig. 2). Interestingly, we found three counties where toads have been collected under both the names DAT and HT (i.e., Harris, Harrison, & Robertson counties; Fig. 2), exemplifying the influence of the date of collection, taxonomic subjectivity of the collector or curator, and the potential for confounding morphology. We also discovered multiple errors in which specimen tags had been misread or contained misleading information. For example, toads collected from Houston, Texas (located in Harris County) were cataloged as toads collected from Houston County, Texas (Table 3). One toad was reportedly collected from “Sam Houston State Park” which can be interpreted as a missense of the two terms “Sam Houston National Forest” with “Huntsville State Park”, because the two co-occur within Walker County, Texas. However, there is also a park in western Louisiana (La Calcasieu Parish) also formerly known by the name “Sam Houston State Park”, although the collector of this particular toad could not recall conducting any work outside of Texas during this time period (T. Matthews, pers. comm.). Dixon (2013) provides species distribution maps by county, based on the existence of museum vouchers, photographs, written accounts, and personal observations. As a consequence, HT is listed as occurring in Washington County, which can be attributed to either a personal observation or simply an error in the presentation of this range map, as no apparent documentation has been found supporting this record and no mention of this county is made in the text (Dixon, 2013). Dixon (2013) also indicates previously occupied counties, now extirpated (i.e., Harris County), with a unique label. Liberty and Fort Bend counties have both harbored well documented populations previously, now thought to be locally extirpated (Yantiss & Price, 1993), ostensibly giving these counties the same status as Harris County, yet these two counties are altogether ignored by Dixon (2013). Interestingly, we did not recover museum vouchers from either of these counties, despite their populations receiving attention from researchers (Yantiss & Price, 1993).

One challenge we repeatedly faced when categorizing physical specimens, or photographs of vouchers, is the prevalence of juveniles or poorly preserved animals. Juvenile Bufonids are difficult, if not impossible, to identify to species. The features reported to parse HT from their allies (i.e., wart size and crests among the head) are not fully formed until adulthood. Further, changes in taxonomic status are often disputed for long periods of time, causing researchers and curators to hesitate, or fail altogether, in changing or updating voucher specimen designations (Pauly, Hillis & Cannatella, 2009). Innovations such as VertNet, as illustrated in our study, aid in identifying these shortcomings so they can begin to be addressed.

Within the counties we examined during this study, there is reportedly a total of eight species of Bufonid (A. a. charlesmithi, A. debilis, A. houstonensis, I. nebulifer, A. punctatus, A. speciosus, A. velatus, A. woodhousii; Dixon, 2013). Yet the status of these eight species is often debated. For example, Tipton et al. (2012) reports controversy concerning the status of DAT as a distinct species (A. charlesmithi, Masta et al., 2002, Fontenot, Makowsky & Chippindale, 2011) rather than a subspecies (Pauly et al., 2004). Additionally, the status of Woodhouse’s Toad (A. woodhousii) has been suggested to include at least two species (A. australis and A. woodhousii; Masta et al., 2002), and two suggested subspecies have previously been suggested (A. w. velatus and A. w. fowleri, Bragg & Sanders, 1951). Both A. w. velatus and A. w. fowleri have previously been accepted as subspecies, (Dixon, 2000; Sanders, 1986), elevated to full species status (A. fowleri and A. velatus, (Fontenot, Makowsky & Chippindale, 2011), or alternatively refuted as unique altogether (Conant & Collins, 1998). Our findings indicate that this may be false with respect to size of post-orbital crests, supporting the remarks of Seal (1994). Additionally, the relative wart size of these animals has not been examined in any quantifiable manner that we are aware of at this time.

While molecular methods (i.e., DNA sequencing) has proven to be a reliable method to differentiate among HT and DAT (Sirsi, Rodriguez & Forstner, 2024), these methods are unlikely to be incorporated into simple studies of herpetofauna inventory or diversity on a small scale. Researchers of other taxonomic groups have successfully vetted their dichotomous keys against the results of DNA sequencing to calculate a level of reliability for traditional methods of identification in the field (Aparicio et al., 2005), and we suggest practitioners and conservation partners consider this approach for the Americanus complex across the Southern United States.

The efficacy of the existing dichotomous keys available for North American bufonids is contingent upon the strength of the findings within the primary literature they draw information from. Small mistakes in foundational literature, such as those identified by Seal (1994), may confound or undermine efforts by researchers, biological consultants, conservation groups, and governing agencies, whose focus may primarily be on enacting protections for endangered species. Our hope is that this study serves as a framework for re-evaluating the morphological, acoustic, and geographic differences between the many sympatric and interbreeding species of Bufonid throughout the South-Central United States.

Conclusions

The motivation for this study stems from decades of disagreement within foundational literature (i.e., species descriptions, dichotomous keys, state and federal agency reports) with respect to the similarities in morphology, male advertisement call, and historic range of these taxa. This is the first article to compare the morphology of HT and DAT quantitatively, as well as the first to examine the historic range for these taxa using museum vouchers. The foundational literature contains many studies that utilize small sample sizes and that lack any form of significance testing. Our study sought to resolve these issues, while revisiting methodologies traditionally used by past researchers. We successfully replicated, and expanded upon, quantitative studies of the morphology of Central Texas Bufonids, and found HT and DAT to overlap considerably in their quantitative morphology. Through the collection of a sufficiently large number of high-fidelity field recordings of wild toads, and analyzing four acoustic characteristics, we resolve previous contradictory accounts regarding male advertisement call and provide context for the differences we discovered. Finally, our examination of vouchered specimens representing these taxa revealed range expansions (varied through time) that had failed to be formally reported.

Our findings emphasize the need for standardized quantitative research into the physical characteristics used to differentiate Bufonids in North America. More specifically, this type of research is imperative within the state of Texas, where sympatry and hybridization confound our current methods of species identification. Given the difficulty in resolving Bufonids of Texas to a specific epithet based on physical characteristics, correctly identifying vouchers within museum collections, as well as those encountered in the wild, is most likely more reliable when examined using molecular techniques (i.e., DNA sequencing)—although these methods were beyond the scope and budget of our study, and due to limitations for DNA extraction from formalin preserved vouchers is not always possible (Janecka, Adamczyk & Gasińska, 2015). While this study has successfully shown that similarities, and differences, exist between these two closely related taxa, we cannot speculate on their shared evolutionary history, divergence, or current status as legitimate epithets, based on the data we examine here.

Supplemental Information

Supplemental Information 1 R code

Script and analysis and figure preparation for both our morphological study and advertisement call comparison

Supplemental Information 2 Advertisement Call Dataset

Acoustic characteristics measured for the Houston Toad and Dwarf American Toad using software Raven.

Supplemental Information 3 Toad Morphology dataset

Measurements of eight morphological characteristics.

We thank the numerous museums and natural history collections, and their staff, for their collaboration, Chris McAllister for agreeing to record DAT vocalizations on our behalf, the Capitol Area Council-Boy Scouts of America for their cooperation and site access, as well as Andy Royle, Ben Bolker, Floyd Weckerly, Rebecca Brunner, as well as two anonymous reviewers, for evaluating earlier drafts of this work.

Additional Information and Declarations

Competing Interests

Author Contributions

Animal Ethics

Field Study Permissions

Data Availability

Andrew R. MacLaren is employed by Cambrian Environmental.

Andrew R. MacLaren conceived and designed the experiments, performed the experiments, analyzed the data, prepared figures and/or tables, authored or reviewed drafts of the article, and approved the final draft.

Toby J. Hibbitts conceived and designed the experiments, authored or reviewed drafts of the article, and approved the final draft.

Michael R.J. Forstner conceived and designed the experiments, authored or reviewed drafts of the article, and approved the final draft.

Shawn F. McCracken conceived and designed the experiments, authored or reviewed drafts of the article, and approved the final draft.

The following information was supplied relating to ethical approvals (i.e., approving body and any reference numbers):

As instructed by the Publishing Editor, here is language regarding our use of passive recording equipment:

Passive recordings were collected with permission from landowners, but did not require any formal permitting from state, federal, or local authorities, and we did not capture any incidental recordings of non-participating human subjects that may have been unaware of the presence of our recording devices.

These statements have been included in our updated manuscript.

The following information was supplied relating to field study approvals (i.e., approving body and any reference numbers):

Access to field sites in Robertson County, Texas, were granted by Laredo Petroleum to Michael R.J. Forstner. Access to field sites (the Griffith League Ranch) in Bastrop County Texas was granted by the Boy Scouts of America Capitol Area Council to Michael R.J. Forstner. Audio Recordings made by C. McAllister in Oklahoma occurred on public property.

No protected areas were accessed during this study, and we do ask that exact locations not be provided to readers for the purpose of protecting sensitive populations.

The following information was supplied regarding data availability:

The raw morphological data and the raw audio characteristic data are available in the Supplemental Files.

The audio is also available at Zenodo: MacLaren, A. (2024). Houston Toad and Dwarf American Toad Supplemental Audio. Zenodo. https://doi.org/10.5281/zenodo.11107152.

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
