# Peer review of "Morphology and vocalization comparison of the Houston Toad and the Dwarf American Toad: implications for their historic range"

_PeerJ, doi:10.7717/peerj.17635_

## Round 0.1 · original submission · Major Revisions

You provide interesting new data on vocal behavior and morphology of Houston and Dwarf American toads which might have important implications for the historic range and conservation efforts of these species. The reviewers were quite split in their assessment. The subject editor did not agree with the assessment of the handling editor and PeerJ re-assigned your submission to me. I apologize for the delayed decision. Based on the points raised by the reviewers as well as my own personal assignment I see no reason to reject your contribution, however I feel several crucial points which need to be addressed before publication. The main points being:

Title: I feel adding something in the title about historical range “estimates” might be appropriate as your methods do not seem to be able to estimate historical range of these species but have important implications for historical “range estimates”.

Main aims: I agree with reviewer 1 and 2 that your goals/aims needs to be more clearly including how the differentiation of two species can contribute to conservations efforts (lines 47-69; compare reviewer 1). Also, how those goals/aims could be addressed or can be reach in the future should be revisiting in the discussion.

Experimental design and method section: the experimental design and the rationale behind it needs to be more clearly described (compare reviewer 1). The method section should focus on methodology and would benefit from additional documentation in the form of a figure and/or table for the anatomical measures highlighted by both reviewers. Please provide example recordings for both species and potentially issues of sound quality (compare reviewer 1). I agree with reviewer 1 that some aspects currently discussed here would better fit in the Introduction or Discussion.

Vocalization analysis: Why was discriminant analysis not used to differentiate between the toad calls (compare reviewer 2) – it would make the approaches more comparable with the morphological analysis. Also, how could intraspecific variation impact on your sampling of two sites for one species and one site for the other species. I agree with reviewer 2 that the caveat should be highlighted more clearly.

Unresolved Museum Records: please reorganized its results in a table as suggested by reviewer 2.

Formatting and structure: please make sure that description of methods, results (focused on new data/studies you performed) and discussion (including interpretations and comparisons with other studies) are clearly separated (compare reviewer 1).

Implications and future perspectives: you find clear similarities in morphology and vocal behavior, but its implications should be more clearly highlighted and ideally also how future studies could address these issues (e.g., potential use of AI instead of discriminant analysis; spatial analyses: compare reviewer 2) . I agree with reviewer 1 that it need to be stated more clearly how your findings contribute to the initially stated conservation efforts. Potentially the the discussion on molecular methods can also be removed unless it can be discussed how they can contribute to reaching one of the goals mentioned in the introduction not possible with other methods you used here (also compare reviewer 2 which even states mentioning these methods earlier). I agree with reviewer 2 that the results are not well aligned with the objectives in the abstract and rewording them would be necessary to match those objectives.

Please make sure to address these as well as all others pointed raised in the revision.

I look forward to receiving your revised manuscript.

Reviewer 1 ·

Basic reporting

MacLaren et al. investigate differences in vocal behavior and morphological parameters between Housten toads and Dwarf American toads. They use acoustic monitoring data from two locations in Texas, two collections of pickled specimens as well as photographs of museum specimens. The goal of the study was to evaluate records of the distribution of the two species. The authors found similarities in morphology and vocal behavior.

Main concerns
The manuscript is lengthy and not very focused in some places. For example, the Introduction states in lines 102 – 103 three goals. How does the differentiation of two species contribute to conservation efforts outlined in lines 47-69? It remains unclear what this investigation will contribute to the field.

Experimental design

It is difficult to evaluate the experimental design because the descriptions are not entirely clear. The Method section is very lengthy and confusing. Some statements seem not to belong in the Methods but in the Introduction or Discussion (e.g.,: “Hillis et al. (1984) provided a quantitative methodology to diagnose species assignment as well as hybrid status for HT and two sympatric species (previously considered congeners) Woodhouse’s Toad (A. woodhousii) and the Coastal Plains Toad (Incilius nebulifer).” line 112-114). It is suggested to thoroughly revise the Method section.

Anatomical measures would probably be easier to understand when in a table and exemplified in a photograph.

The sound recording might not be of sufficient quality. Figure 4 shows multiple sounds. Some explanation is required. Maybe add example recordings from both species.

The Result section is a mix of methods, results, and discussion. A thorough revision is necessary.

Validity of the findings

The discussion section is mostly a continued literature review rather than a discussion of the results. Maybe start the discussion with a summary of the main results and then discuss each result. It seems also important to revisit the question of how your findings contribute to the initially stated conservation efforts. Maybe also remove the discussion on molecular methods since this seems not relevant here.

The authors claim there are not differences between the two toad species. What does that mean and what are the implications for any conservation efforts?

The authors claim that "alternatively" (Line 406) sound recordings could differentiate the two species. I do not see that in the results. This seems to be speculation. Please clarify.

Reviewer 2 ·

Basic reporting

This is a very clear, well written, pleasant to read, article. The objectives are clearly stated, and the entire article flows logically.

I only have minor suggestions:
1) Lines 39-44. The results in the abstract are not well aligned with the objectives. I suggest rewording so it matches the three objectives. I also suggest the authors add a conclusion sentence.

2) Lines 384-386. This is the first time that authors mention that genetics has been done. I suggest the authors state this earlier, preferably in the Introduction. Then, in the Discussion, I suggest the authors provide more information about the genetic findings.

3) I suggest the authors add a schematic figure of a frog, depicting all the morphological characters assessed.

Experimental design

Experimental design is generally appropriate. I have several suggestions though.

1) The authors could not differentiate between the two toads using a discriminant analysis. Recent work has been using AI for this purpose. If authors agree, I suggest mentioning that AI could be used in the future.

2) The authors used a t-test to test for differences between the toad call. Could discriminant analysis be used here? If so, I suggest authors consider it.

Validity of the findings

The authors provide all raw data and findings are sound. I really like the conclusions. I only have a few minor comments:

1) In the vocalization analysis, the authors do not account for intraspecific variation since they only sampled two sites for one species and one site for the second species. I suggest the authors clearly state that as a caveat.

2) The results section for the Unresolved Museum Records is chaotic and difficult to follow. Can authors summarize the results in a table?

3) The authors have not analyzed any of the data with respect to geography. Could there be a geographic cline that would allow a better discrimination between the two species? For example, some closely related species display character displacement in sympatry or parapatry. If authors don't feel they have enough data to perform their analyses in a spatial context, I suggest they mention this possibility.

---

## Round 0.2 · accepted · Accept

Thank you for addressing our suggestions. The manuscript is now easier to following, the aims are more clearly described and the implications also more easier to reproduce. I agree with the assessment of reviewer 1 that the manuscript can known be accepted pending two small issues are corrected during proofing phase. Please make sure one typographical issue is correct (line 305: "the8orphologyical" should be "the morphological") and consider using "based on" instead of "on the basis of" (lines 31, 146).

Reviewer 1 ·

Basic reporting

no comment

Experimental design

no comment

Validity of the findings

no comment

Additional comments

The authors made an effort to address my concerns regarding the last version of the paper.
The present version provides an improved overall understanding of how the differentiation of two species can contribute to conservation efforts.